# Dynamics of the formation of flat clathrin lattices in response to growth factor stimulus

Lingxia Qiao[1,2], Marco A. Alfonzo-Méndez[3], Justin W. Taraska[3], Padmini Rangamani[2,4]*

1 School of Data Science, Fudan University, Shanghai, China, 2 Department of Pharmacology, School of Medicine, University of California San Diego, San Diego, California, United States of America, 3 Biochemistry and Biophysics Center, National Heart, Lung, and Blood Institute, National Institutes of Health, Bethesda, Maryland, United States of America, 4 Department of Mechanical and Aerospace Engineering, Jacob's School of Engineering, University of California San Diego, San Diego, California, United States of America

* prangamani@health.ucsd.edu

## Abstract

Clathrin assemblies on the cell membrane are critical for endocytosis and signal transduction in cells. Specifically, the Ω-shaped clathrin assembly functions as the coat of endocytic vesicles, while the flat clathrin assembly, also known as the flat clathrin lattice (FCL), serves as a signaling hub for various pathways. Multiple flat clathrin lattices exist on the cell membrane, and these lattices grow after epidermal growth factor stimulation (EGF) and then return to baseline. In this work, we used a particle-based model to simulate the assembly and disassembly of flat clathrin lattices to capture these dynamics. We found that the formation of flat clathrin lattices is highly dynamic, that is, cluster number, size and dwelling time often change even in the absence of any stimulus. Moreover, these key features are affected by adaptor protein 2 (AP-2) number, clathrin-clathrin binding rate, and clathrin diffusion coefficient. Specifically, an increase in AP-2 number leads to the transition from no cluster, short-lived multiple small clusters, to a long-lasting single giant cluster. An increased clathrin-clathrin binding rate or decreased clathrin diffusion coefficient both result in an increased cluster number, reduced cluster size, and shortened dwelling time. Furthermore, we also predicted that under EGF stimulation, simultaneous changes in the AP-2 number, the clathrin-clathrin binding rate, and the clathrin diffusion coefficient can reproduce the experimentally observed trend of FCLs: an increase in cluster number and size in the first 30 minutes, followed by a decrease after 30 minutes. These findings reveal kinetic mechanisms underlying the formation of multiple FCLs and how EGF regulates FCL dynamics.

**Data availability statement:** The executable input files for the models can be found in https://github.com/RangamaniLabUCSD/FlatClathrinLattice.

**Funding:** This work was supported by NIH R01GM132106 (to PR), NIH NIGMS R35GM158446(to PR), Army Research Office W911NF2310249(to PR), and Intramural Research Program of the National Heart, Lung, and Blood Institute, National Institutes of Health, USA(to JWT). The funders had no role in study design, data collection and analysis, decision to publish, or preparation of the manuscript.

**Competing interests:** The authors have declared that no competing interests exist.

## Author summary

Clathrin is a protein essential for cell processes like endocytosis and signaling. It can form two shapes: one that helps create vesicles, and another, the "flat clathrin lattice" (FCL), which is involved in cellular signaling. In this study, we simulated how FCLs assemble and disassemble on the cell membrane. We found that FCLs are highly dynamic, with their size, number, and lifespan fluctuating even without external signals. Key factors like adaptor protein 2 (AP-2) and clathrin-clathrin interactions influence these dynamics. Our model also predicted that, under EGF stimulation, simultaneous changes in the AP-2 number, the clathrin-clathrin binding rate, and the clathrin diffusion coefficient can reproduce the experimentally observed trend of FCLs. These findings provide insight into how cells regulate FCL formation and signaling in response to external stimuli.

## 1 Introduction

Clathrin, a protein shaped like a triskelion, composed of three clathrin heavy chains and three light chains, plays an important role in growth factor signaling, adhesion, and endocytosis [1,2]. Clathrin assembles into organized clusters on the cell membrane. One widely observed shape of clathrin assembles is the $\Omega$-shape on the plasma membrane (PM) during endocytosis [3]. During endocytosis, clathrin molecules are recruited to the plasma membrane (PM) by adaptor protein complex 2 (AP-2), a major clathrin-associated adaptor [4]. Subsequently, the associated membrane bends into $\Omega$-shaped pits [2]. In addition to the classic $\Omega$ shape pits, clathrin can also form a flat lattice structure on the PM (also termed plaques) [5]. Unlike $\Omega$-shaped clathrin assemblies, flat clathrin lattices (FCLs) are long-lived stable structures [6, 7]. Recent studies have revealed that FCLs play an important role in cell functions. FCLs are consistently associated with actin [8] and have been shown to work with actin to oppose cell migration and contribute to skeletal muscle sarcomere organization [9,10]. In addition, FCLs are enriched with $\beta5$-integrin [10–12], a receptor that plays a key role in cell proliferation and cell adhesion [13,14]. Moreover, FCLs also regulate cell signaling by interacting with several signaling pathways, such as the epidermal growth factor (EGF), AKT, and hepatocyte growth factor pathways [9,15–17].

Many mathematical models have been widely used to explore the mechanism of $\Omega$-shaped clathrin in the absence or presence of cell membranes. In the absence of cell membranes, clathrins can still form a cage shape (the closed geometry of $\Omega$ shape) [18], and the clathrin is usually modeled as a coarse-grained triskelion particle whose interactions are controlled by the potential energy [19–22]. However, clathrins closely interact with the cell membrane through adaptor proteins to form the $\Omega$-shaped pits during endocytosis, thus inspiring the models that incorporate the cell membrane to the $\Omega$-shaped clathrin [1,4,23–25]. For systems focusing on the biochemical aspects of the cell membrane, such as lipid localization and adaptor recruitment, clathrin is usually modeled as a coarse-grained triskelion particle, and

computational approaches include Brownian dynamics with potential-based interactions [18,26–28] and single-particle reaction-diffusion models [29–31]. For systems focusing on the biophysical aspects of cell membrane (e.g., tension and bending rigidity), clathrin is simplified to a zero-volume dot, and continuum field-based mesoscale models are preferred [24,25,32–34]. The involvement of the cell membrane opens the possibility to answer broader physiologically relevant questions such as the role of adaptors, cytoskeleton, and volume-to-area ratio in Ω-shaped clathrin, and the effect of clathrin dynamics on endocytosis.

In addition to clathrin, interactions between other molecules and cell membranes have also been extensively investigated using models that span multiple scales, from individual molecules at the nanometer level to small membrane patches at the micrometer level [35,36]. For example, molecular dynamics (MD) simulations, which are widely used to capture conformational changes in tumor necrosis factor (TNF) and its receptor [37,38], have recently been integrated with a domain-based coarse-grained diffusion–reaction model to examine receptor clustering on the membrane [39]. Ying et al. used a continuum membrane mechanics model coupled to stochastic rigid-body simulations of HIV-1 Gag lattice assembly to quantify the membrane bending energy for different lattice eccentricity [40]. In another example, Guan et al. employed a two-dimensional particle-based model with a zero-volume dot assumption to reveal the role of feedback loops in polarity site formation during yeast mating [41].

Most of the models for Ω-shaped clathrin can be modified to models for the formation of FCL including particle-based molecular dynamics or Brownian dynamics with potential-based interactions [42], single-particle reaction-diffusion models [30,43]. By changing the clathrin rotation [43] or strain energy [30], the simulated shape of clathrin can vary between Ω and lattice structure. Another way to acquire the clathrin lattice is to constrain the simulation domain to 2D [42,44]. Furthermore, the 2D simulation domain can be discretized to a hexagonal lattice, and one clathrin occupies six edges of two neighboring hexagons, i.e., the lattice model [44]. Through simulations of these models, the formation of FCL has been reproduced, and the role of kinetics, for example, adaptor stoichiometry, clathrin concentration, and volume-to-area ratio, have been demonstrated [30,42–44].

The computational studies mentioned above primarily focused on a single FCL. However, multiple FCLs exist on the cell membrane (Fig 1A). The distance between the two nearest FCLs can vary, as shown by the various distances between two FCLs in Fig 1B. The mean size of FCL in HeLa and U87 cells is smaller than 0.1 μm² (Fig 1C). Furthermore, for each FCL, there usually exists another FCL in the neighboring 1 μm × 1 μm square (Fig 1D). Compared with HeLa and U87 cells, MCF7 cells exhibit a larger number and size of FCLs on the cell membrane (S1 Fig). Thus, FCLs are ubiquitous, having been observed in multiple cells from different origins [17,45]. However, it remains unclear how neighboring multiple FCLs can co-exist instead of merging into one big cluster.

Moreover, multiple FCLs exhibit dynamic changes under EGF stimulus. Under the stimulus of the epidermal growth factor (EGF), FCLs on cellular membranes have been observed to grow in size and number, followed by a decrease toward baseline by 60 min (Fig 1E and [16]). More precisely, in the first 15 minutes after the stimulus of EGF, the number of FCLs increases nearly 3.5-fold (from 40 to 140), and maximal FCL area on the cell membrane grows near 6-fold (from 0.05 μm² to 0.3 μm²). It should be noted that the increase in the size and number of FCLs occurs simultaneously. Such trend for cluster size and number also holds for the neighboring 1 μm × 1 μm square areas around each cluster (panel D in S1 Fig). During this process, EGFR, scaffold protein Grb2, and tyrosine kinase Src are all recruited to FCLs [16]. However, the underlying mechanism of such FCL dynamic changes remains unclear.

The above discoveries of multiple FCLs and the dynamics of EGF-triggered FCLs raise interesting questions about the biochemical mechanisms that govern the interaction between clathrins and the cell membrane. For example, how does clathrin form multiple stable clusters? What changes to kinetic parameters are required after EGF stimulation to achieve the observed behavior, that is, an initial increase in both the number and size of clusters in the first 30 minutes, followed by a return to baseline levels? Here, we sought to answer these questions using computational modeling.

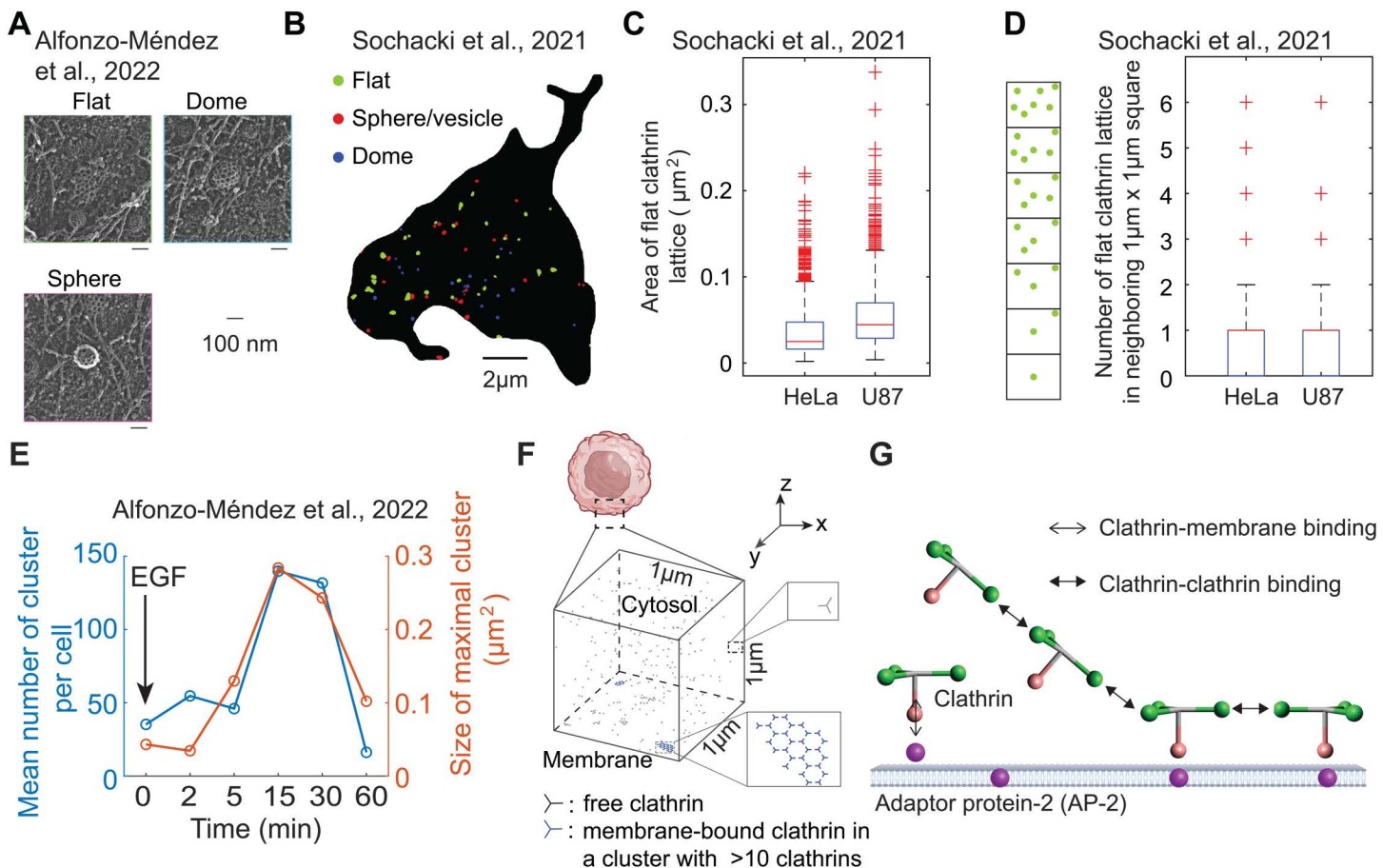

**Fig 1. Multiple flat clathrin lattices (FCLs) coexist and undergo dynamic change under EGF stimulus. (A)** A representative crop from a the montaged platinum replica transmission electron microscopy (PREM) image of an unroofed cell. This panel was adopted from Supplementary Figure 1a in [16] (CC BY 4.0). **(B)** Clathrin masks based on the PREM image of an unroofed HeLa cell. Clathrin clusters in flat, spherical, and dome shapes are indicated by green, red, and blue, respectively. Scale bar: 2 µm. Data are from [45]. **(C)** Area of flat clathrin lattices for HeLa cells (N = 18) and U87 cells (N = 12). In each box plot, the central red line denotes the median; bottom and top black edges indicate the 25th and 75th percentiles, respectively; whiskers denote the minimum and maximum extreme data points; plus markers represent outliers. Data are from [45]. **(D)** Number of FCLs in neighboring 1 µm × 1 µm square. We used each FCL as a center and then calculated the number of FCLs in the neighboring 1 µm × 1 µm square. The box plots were plotted in the same way as in **(C)**. The data is based on the same HeLa cells (N = 18) and U87 cells (N = 12) as those in **(C)**. **(E)** The total number of all FCLs (left axis) and size of maximal FCL (right axis) as a function of time under EGF stimulus. Figures are based on the data in [16]. **(F)** The cubic simulation domain. The bottom surface denotes membrane, and the area above denotes the cytosol. The cell cartoon was created in BioRender. Rangamani, **P.** (2026). https://BioRender.com/mfs2811. **(G)** Particle-based model that describes the binding and unbinding events of adaptor protein 2 (AP-2) and clathrins. This model is similar to that in [30]. The AP-2 (purple sphere) is located on the cell membrane. Clathrin (particle composed of one pink head and three green legs) can bind to AP-2 or other clathrins.

## 2 Models and methods

In order to model the structure of flat clathrin lattice explicitly, we first chose a cube with edge length 1 µm (one-twentieth of the diameter of HeLa cell [46]) as the simulation domain (Fig 1F). The bottom surface denotes a patch of the cell membrane and the volume above denotes the cytosol close to this membrane patch. Thus, we are able to simulate the exchange of clathrins between cell membrane and cytosol.

Since AP-2 is involved in the formation of FCLs by recruiting clathrin from the cytosol to the cell membrane, we included AP-2 in our simulation. Thus, we considered the diffusion and biochemical interactions of both clathrin and AP-2.

AP-2 and clathrin are assumed to be located on the cell membrane and in the cytosol, respectively. However, clathrin can bind to the cell membrane by binding to AP-2 through the AP-2 binding site on clathrin. Moreover, to achieve rapid clathrin-clathrin binding near the membrane, the binding rate is modeled with two distinct values depending on whether at least one clathrin is bound to AP-2: the rate is set to zero if neither interacting clathrin is bound to AP-2, and non-zero if at least one of them is bound to AP-2. Note that, we set the clathrin–clathrin binding rate to zero (instead of the 0.083 $\mu$M$^{-1}$s$^{-1}$ reported in [30]) when neither clathrin is bound to AP-2, as this results in a higher number of clusters compared to a non-zero binding rate (S2 Fig). Thus, AP-2 bound clathrin can recruit other clathrins in the cytosol once these two clathrins are close enough, which allows the emergence of clathrin clusters. All of these association events are reversible. Kinetic parameters are from [30] (also see S1 Table), which were optimized to fit the fold change of clathrin cluster size in *in vitro* experiments [47].

In addition to biochemical reactions, individual clathrin, individual AP-2 and assembled cluster can all undergo both translational and rotational diffusion. Traditionally, diffusion coefficients are measured by labeling particles of interest with fluorescent proteins, monitoring their fluorescence (e.g., via fluorescence recovery after photobleaching [48]), and extracting diffusion coefficients from the experimental data using mathematical models [49]. However, fluorescence labeling strategies and mathematical models may vary depending on the particle and its surrounding environment [50–55]. In this study, we used diffusion coefficients reported in [30]. In detail, the translational diffusion coefficients of molecules $i$ ($i$ = AP-2, Clat or cluster) are denoted as ($D_{i,x}$, $D_{i,y}$, $D_{i,z}$), where each component corresponds to diffusion along the $x$-, $y$-, and $z$-directions, respectively. Similarly, the rotational diffusion coefficients are denoted as ($D_{R,i,x}$, $D_{R,i,y}$, $D_{R,i,z}$), where $x$, $y$, and $z$ indicate the axis of rotation. For individual clathrin or individual AP-2, the translational and rotational diffusion coefficients are from [30] (also see S1 Table), which were obtained using Stokes-Einstein equation as shown below:
$D_{AP\text{-}2,x} = D_{AP\text{-}2,y} = \frac{k_B T}{6\pi\eta r_{AP\text{-}2}}$, $D_{R,AP\text{-}2,z} = \frac{k_B T}{8\pi\eta r_{AP\text{-}2}^3}$. $D_{Clat,x} = D_{Clat,y} = D_{Clat,z} = \frac{k_B T}{6\pi\eta r_{Clat}}$, $D_{R,Clat,z} = D_{R,Clat,y} = D_{R,Clat,z} = \frac{k_B T}{8\pi\eta r_{Clat}^3}$. Here, $k_B$ is the Boltzmann constant, $T$ is the absolute temperature, $\eta$ is the viscosity of the medium (assumed to be water), and $r$ is the hydrodynamic radius of the particle. Moreover, the values of $D_{AP\text{-}2,z}$, $D_{R,AP\text{-}2,x}$, and $D_{R,AP\text{-}2,y}$ are set to zero, thereby constraining AP-2 molecules to remain on the cell membrane, as the cell membrane is parallel to the $x-y$ plane (Fig 1F). The settings for $D_{AP\text{-}2,z}$, $D_{R,AP\text{-}2,x}$, and $D_{R,AP\text{-}2,y}$ also imply that a particle can have different effective hydrodynamic radii for translational and rotational diffusion, as well as for distinct directions.

Once AP-2 binds to clathrin and forms a cluster, the assembled cluster is regarded as an entire unit to diffuse, whose diffusion coefficient is calculated as below. In general, according to the Stokes-Einstein equations, given the translational and rotational diffusion coefficients $D$ and $D_R$ of a particle, the corresponding effective radii for translation and rotation can be written as $\frac{k_B T}{6\pi\eta}D^{-1}$, $\frac{k_B T}{8\pi\eta}D_R^{-1/3}$, respectively. Thus, by assuming that the radius of the cluster is the summation of radius of all component (i.e., $r_{\text{cluster}} = \sum_{i\in\text{cluster}} r_i$), the radius of the cluster can be expressed as $\sum_{i\in\text{cluster}} \frac{k_B T}{6\pi\eta}D_i^{-1}$ or $\sum_{i\in\text{cluster}} \frac{k_B T}{8\pi\eta}D_{R,i}^{-\frac{1}{3}}$. Next, according to Stokes-Einstein equation, the translational diffusion coefficient of the cluster is $D_{\text{Cluster}} = \frac{k_B T}{6\pi\eta r_{\text{cluster}}} = \frac{k_B T}{6\pi\eta \sum_{i\in\text{cluster}} \frac{k_B T}{6\pi\eta}D_i^{-1}} = \left[\sum_{i\in\text{cluster}} D_i^{-1}\right]^{-1}$. Similarly, the rotational diffusion coefficient of the cluster is $D_{R,\text{Cluster}} = \frac{k_B T}{8\pi\eta r_{\text{cluster}}^3} = \frac{k_B T}{8\pi\eta \left[\sum_{i\in\text{cluster}} \frac{k_B T}{8\pi\eta}D_{R,i}^{-\frac{1}{3}}\right]^3} = \left[\sum_{i\in\text{cluster}} D_{R,i}^{-\frac{1}{3}}\right]^{-3}$. Therefore, by applying $D_{\text{Cluster}} = \left[\sum_{i\in\text{cluster}} D_i^{-1}\right]^{-1}$ and

$D_{R,\text{Cluster}} = \left[\sum_{i\in\text{cluster}} D_{R,i}^{-\frac{1}{3}}\right]^{-3}$ to directions $x$, $y$ and $z$ and to different types of diffusion, the diffusion coefficients of the assembled cluster can be calculated. These equations for diffusion coefficients of clusters have also been used in [30]. Note that, clathrins can only form clusters by binding to AP-2, which later triggers the binding to other clathrins. Therefore, every clathrin cluster contains at least one AP-2 molecule. Combined with the values of $D_{AP\text{-}2,z}$, $D_{R,AP\text{-}2,x}$ and $D_{R,AP\text{-}2,y}$, we have $D_{Cluster,z} = 0$, $D_{R,Cluster,x} = D_{R,Cluster,y} = 0$, meaning that the cluster cannot diffuse out of the membrane [56].

Next, we describe how clathrin and AP-2 are modeled. Each clathrin is modeled by a rigid structure (in green and pink in Fig 1G) that can diffuse in the cytosol or on the membrane. For each clathrin, there are four binding sites (see S2 Table

for the length of binding sites): three binding sites (green in Fig 1G) that allow it to bind to another clathrin, and one binding site (pink in Fig 1G) that allows clathrin to bind to AP-2. This structure of clathrin is the same as that in [30] except that the binding site for AP-2 is decreased to 1. This simplification is used to reduce computation time, and we assume that it does not quantitatively affect the results. Moreover, the clathrin-clathrin binding event is assumed to be achieved by head to head binding of clathrin binding sites from two different clathrins. After binding, the distance between two bound sites is 5 nm, and the center of two clathrins and all clathrin binding sites are on the same plane. As for AP-2, it is represented as a zero-volume dot that is always located on the cell membrane. Although all AP-2 molecules are located on the membrane, they exist in two forms: clathrin-bound and clathrin-unbound. The location of clathrin-bound AP-2 is determined by the clathrin binding site, while clathrin-unbound AP-2 molecules are not explicitly modeled (S3 Fig). This implicit localization assumption for AP-2 is based on the *in vitro* model described in [30]. For the binding of clathrin to AP-2, the AP-2 binding site on the clathrin is oriented perpendicular to the cell membrane. The excluded volume of clathrin is set at 10 nm. These settings ensure the formation of a flat hexagonal structure of clathrins on the cell membrane.

Then, to model the dynamics of clathrins in the cubic simulation domain, we utilized the particle-based algorithm NERDSS, a nonequilibrium simulator for multibody self-assembly developed by Varga et al. [31]. The time step $\Delta t$ is set to 3 μs, during which we assumed that each clathrin either diffuses or undergoes only one reaction. Within each time step, the dissociation events are tested first. The corresponding reaction probability of the dissociation event is calculated as $1 - exp(-k\Delta t)$ from a Poisson process, where $k$ is the dissociation rate with the unit $s^{-1}$. Then, the association events are tested, where the reaction probability is calculated by free propagator reweighting (FPR) algorithm [31,43,57]. If the particle undergoes dissociation or association events, this particle cannot diffuse in this time step. Therefore, we only consider the diffusion of those molecules that are not involved in dissociation or association events in this time step. For translational diffusion, the displacement of a single molecule or clathrin cluster along direction $p$ ($p = x, y, z$) is given by $\Delta p = \sqrt{2D_p\Delta t}\xi_p$, where $\xi_p$ is a random number from a standard normal distribution, and $D_p$ denotes the translational diffusion coefficient along the corresponding direction. For the rotational diffusion process, the single molecule or clathrin cluster are set to rotate around the $p$ axis ($p = x, y, z$) with the angle $\sqrt{2D_{R,p}\Delta t}\eta_p$, where $\eta_p$ is also a random number from a standard normal distribution, and $D_{R,p}$ denotes the rotational diffusion coefficient around the corresponding axis. At the same time, the algorithm checks whether the newly displaced positions will cause overlapping of the clathrin molecules. If overlap occurs, the displacement is resampled until non-overlapping positions among the clathrins are achieved.

The simulation domain is bounded by six surfaces (two along each of the x, y, and z axes). All these six surfaces are set as reflecting boundaries: if the tentative next position of a clathrin or clathrin cluster is outside the domain, it is reflected back inside. In detail, let the current position of a molecule be denoted as $(pos_1, pos_2, pos_3)$, corresponding to its coordinates along the $x$, $y$, and $z$ axes. Reactions or diffusion may move the molecule to a tentative new position $(pos_1^{new}, pos_2^{new}, pos_3^{new})$. If only $pos_1^{new}$ exceeds the right boundary at $x = L$, the y- and z-coordinates remain unchanged (i.e., $pos_2^{new}$ and $pos_3^{new}$), but the x-coordinate is reflected back into the domain and given as $2L - pos_1^{new}$. However, the association event is rejected if it can cause the clathrin cluster to extend beyond the simulation domain.

The input files used for NERDSS are available at https://github.com/RangamaniLabUCSD/FlatClathrinLattice. The simulations were performed at Triton Shared Computing Cluster at the San Diego Supercomputer Center (https://doi.org/10.57873/T34W2R).

## 3 Results

In this study, we investigated the underlying mechanisms of two phenomena: (1) the formation of multiple FCLs in the absence of stimulation, and (2) the EGF-induced dynamics of FCLs, characterized by an increase in both their size and number, followed by a return to baseline levels. We utilized NERDSS, a nonequilibrium reaction-diffusion self-assembly simulator [30], to describe the assembly and disassembly of clathrins. In this model, each clathrin is an individual particle and interacts with each other through binding and unbinding events (Fig 1F–1G). We found that even if all the kinetic

parameters remain unchanged, FCLs are very dynamic, meaning the number and size of FCLs often vary with time. The numbers of FCLs, size of each FCL, and dwelling time are affected by adaptor protein-2 (AP-2) number, clathrin-clathrin binding rate, and clathrin diffusion coefficient. Furthermore, simultaneous changes in AP-2 number, clathrin-clathrin binding rate, and clathrin diffusion coefficient can generate the experimentally observed FCLs dynamics triggered by EGF, that is, the increasing trend of cluster number and size in the first 30 minutes and then decreasing back to baseline levels. These findings reveal the formation mechanisms of multiple FCLs and provide insights into the mechanisms underlying EGF-triggered FCL dynamics, enhancing our understanding of how FCLs form and change on the cell membrane.

### 3.1  The formation of flat clathrin lattices is dynamic

Given the presence of multiple FCLs in various cell types even without stimulus (Fig 1A–1D and [45]), we first studied the mechanism of spontaneous formation multiple FCLs, corresponding to the case without any stimulus. We set the randomly distributed clathrins as the initial condition (Fig 2A), and then simulated clathrin dynamics. We found that the cluster is always dynamic: there is one cluster at 21 minutes but two clusters at 29.7 minutes (Fig 2B). The mean cluster size, defined as the total cluster size divided by the cluster number, also changes from 0.075 $\mu m^2$ to 0.0285 $\mu m^2$. Such dynamic transitions in cluster number (blue curve in Fig 2C) occur frequently and persists even when the number of membrane-bound clathrins reaches the plateau (red curve in Fig 2C). These results indicate that FCLs on the cell membrane can transition between different states.

To quantitatively measure the FCL dynamics on the cell membrane, we defined three metrics for the most possible pattern: cluster number, mean value of the mean cluster size, and mean value of dwell time. First, we defined $t_c$ as the time when the number of membrane-bound clathrins reaches 90% of that at the end of simulations (i.e., 30 minutes). We chose the time 30 minutes due to the long-lived property of FCL which ranges from 2 to 10 min to more than 1 h [6,11]. We only focused on the behavior after $t_c$ (Fig 2C), because the number of membrane-bound clathrins only shows small fluctuations after $t_c$. Next, for each time point, we calculated the mean cluster size and found that it shows high variation across all time points (panel A in S4 Fig). Therefore, due to the large variations, the mean of the mean cluster size across all time points does not accurately reflect clathrin dynamics. To address this issue, we categorized all data frames into different patterns, which is defined as the sub-set of data frames with the same cluster number. Interestingly, we found that for a given pattern, the variation of the mean cluster size is much smaller than that across all data frames (panels B-G in S4 Fig). Therefore, instead of considering all data frames, we focused on the pattern, that is, the subset of data frames with the same cluster number. Furthermore, we only focused on the pattern with the highest frequency, referred to as the most possible pattern, since it is likely to be observed by the experiment. For example, in the simulation shown in Fig 2B, the most possible pattern has a cluster number of 2 (Fig 2D). Then, we calculated the mean value of the mean cluster size and the dwelling time for the most possible pattern (Fig 2F). Here, the dwelling time is the duration that the cluster number remains unchanged (arrow in Fig 2C). For example, for the simulation in Fig 2B, the most possible pattern has a mean cluster size of 0.03$\mu m^2$ and a dwelling time of 0.69 minutes. Overall, these results suggest that the clathrin cluster exhibits frequent changes in cluster number, size, and dwelling time in the absence of any stimulus.

### 3.2 Number of AP-2 affects the cluster number, size and dwelling time

Previous studies showed that sufficient adaptor proteins are required to maintain the FCL [30], but the effect of AP-2 number on the cluster number, size, and dwell time remains unknown. To investigate this, we simulated the clathrin dynamics for different numbers of AP-2. The initial condition is the same as that in Fig 2A, that is, randomly distributed clathrins. Despite the dynamic transition between different states, the number of clusters is 0, larger than 1, and 1 when AP-2 number is 10, 200, and 400, respectively (Fig 3A–3C). By testing more values of the AP-2 number with 4 replicates for each value, the cluster number for the most possible pattern shows a significant increase and then decrease when the AP-2 number increases (Fig 3D). These results indicate that increasing AP-2 number leads to the transition from no cluster,

PLOS Computational Biology

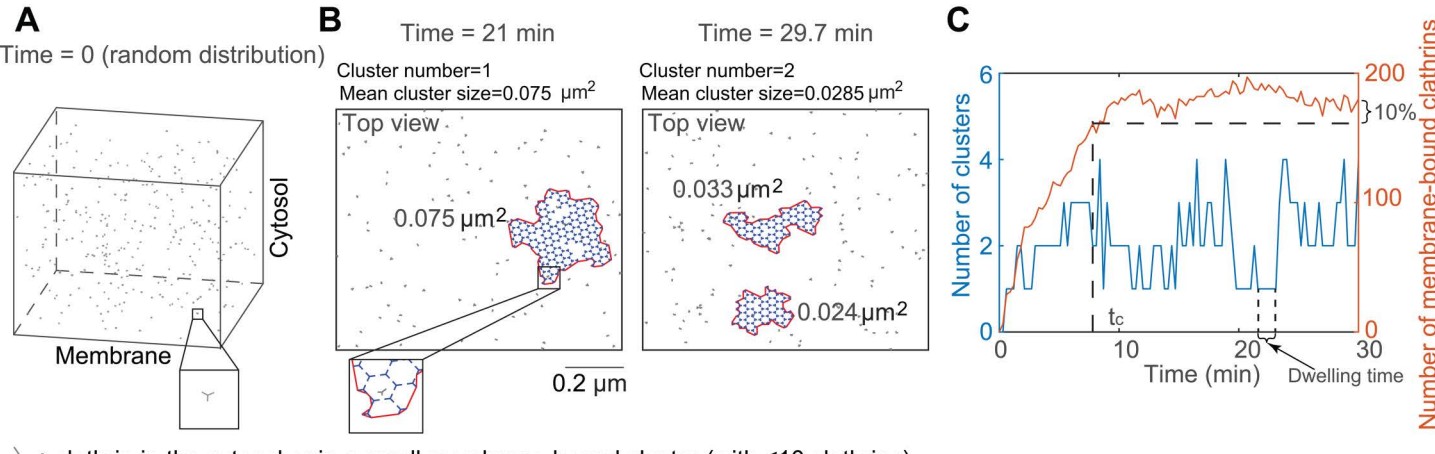

: clathrin in the cytosol or in a small membrane-bound cluster (with ≤10 clathrins)

: clathrin in a large membrane-bound cluster (with >10 clathrins)

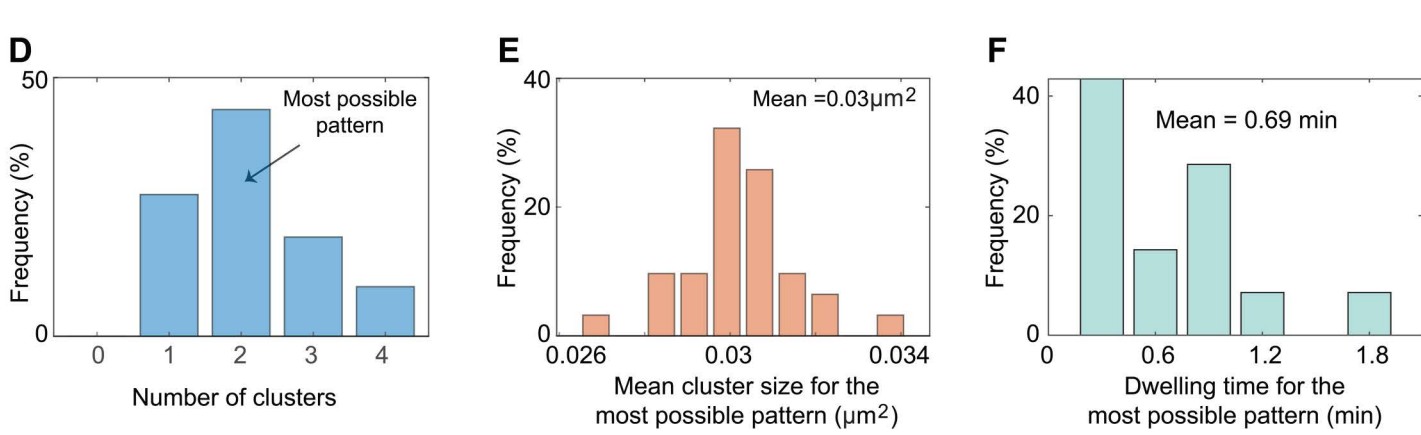

**Fig 2. The FCL formation is highly dynamic. (A)** The initial condition. Clathrin molecules (denoted in gray) are randomly distributed in the simulation domain at time 0. **(B)** Snapshots of FCLs at 21 minutes (left) and 29.7 minutes (right) when simulating from the random initial condition in **(A)**. The gray particle represents the clathrin either in the cytosol or in a small membrane-bound cluster (with no more than 10 clathrins), while the blue particle represents clathrin in a large membrane-bound cluster (containing more than 10 clathrins). The number of AP-2 is set to be 150. The area surrounded by the red curve is calculated as the size of each cluster. Mean cluster size is defined as the total size of all clusters divided by the number of clusters at a specific time point. **(C)** The number of clusters (left axis) and that of membrane-bound clathrins on cell membrane (right axis) as functions of time. The settings of model simulations are the same as that in **(B)**. We defined $t_c$ as the time when the number of clathrins on the cell membrane first exceeds 90% of the amount present at 30 minutes, and assumed that the system after $t_c$ is at equilibrium. **(D)** The frequency of cluster numbers after $t_c$. The cluster number with the highest frequency is defined as the most possible pattern. **(E)** The frequency of mean cluster size for the most possible pattern. **(F)** The frequency of dwelling time for the most possible pattern.

multiple small clusters to one giant cluster. It should be noted that the clathrin-clathrin binding rate we used is 50 times as large as that in [30], as the latter cannot generate the experimentally observed multiple clusters, even when the AP-2 number is varied (panel D in S1 Fig D and S5 Fig). When the clathrin-clathrin binding rate is low, the clathrin behavior with increased AP-2 number is the same as that in [30], that is, a transition directly from no cluster to one giant cluster without experiencing the multiple small clusters stage (S5 Fig). In addition to the effect on cluster number, the AP-2 number also affects the mean cluster size and dwelling time. As the number of AP-2 increases, the mean value of the mean cluster size monotonically increases (Fig 3E). At the same time, the dwelling time for the most possible pattern first decreases and then increases (Fig 3F), which is opposite to the trend of the cluster number. Combining the effects of AP-2 number on

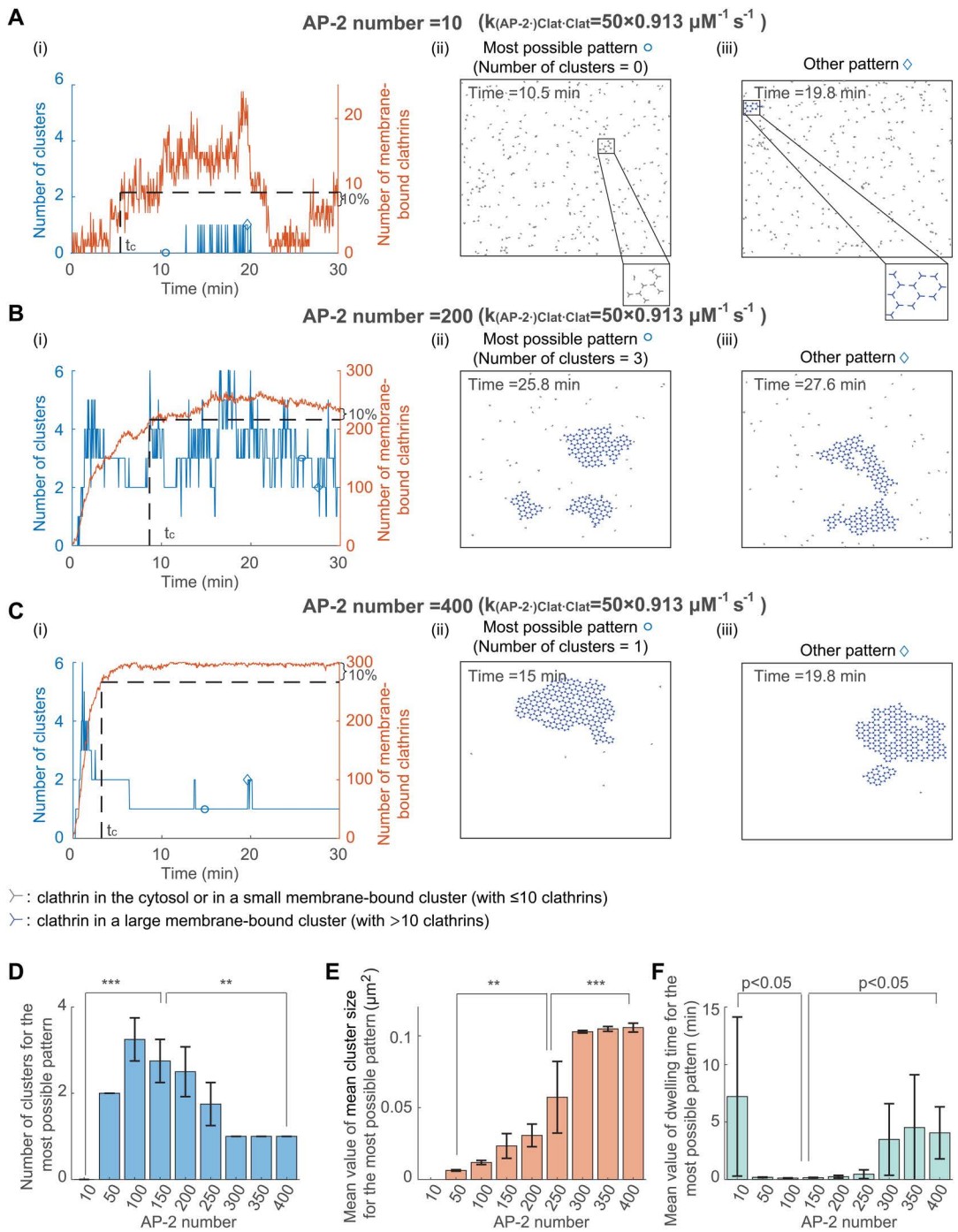

**Fig 3. Number of AP-2 affects the FCLs cluster number, size and dwelling time. (A-C)** Simulated clathrin dynamics when the AP-2 number is 10 **(A)**, 200 **(B)**, and 400 **(C)**. The initial condition is randomly distributed clathrins in the simulation domain. In panel **(i)**, the number of clusters (in blue) and the number of membrane-bound clathrins (in red) are shown as a function of time. Panels (ii) and (iii) show the most possible pattern and the other pattern, respectively. The circle or diamond indicates the time point in the panel **(i)**. **(D)** The cluster number for the most possible pattern when increasing the AP-2 number. **(E)** The mean value of the mean cluster size for the most possible pattern when increasing the AP-2 number. **(F)** The mean dwelling time for the most possible pattern when increasing the AP-2 number. P value was obtained from unpaired t-test: ** indicates $p<0.01$;*** indicates $p<0.001$. Data in (D-F) were shown as mean±SD, where SD means standard deviation.

cluster number, cluster size, and dwell time, we concluded that increasing the number of AP-2 leads to the transition from no cluster to short-lived multiple small clusters, to a long-lasting single giant cluster.

### 3.3 Increasing clathrin-clathrin binding rate leads to an increased cluster number, reduced cluster size, and shortened dwelling time

While only an intermediate value of the AP-2 number leads to the formation of multiple clusters, we next explored whether other parameters can robustly generate multiple clusters. Since the clathrin-clathrin binding rate in [30] was estimated by fitting to clathrin dynamics in a membrane tube, the actual clathrin-clathrin binding rate in real cells may differ from that in [30]. Therefore, we studied the influence of clathrin-clathrin binding rate on cluster formation. Throughout this work, the binding and unbinding rates refer to the association and dissociation rates, respectively. We changed the value of clathrin-clathrin binding rate $k_{(AP-2\cdot)Clat\cdot Clat}$ while fixing the value of other kinetic parameters. The initial condition is still a random distribution of clathrins. As the clathrin-clathrin binding rate $k_{(AP-2\cdot)Clat\cdot Clat}$ increases, the most possible pattern shows increased cluster number (Fig 4A–4D), decreased mean value of the mean cluster size (Fig 4E), and decreased mean dwelling time (Fig 4F). Note that the third bar in Fig 4D is identical to the third bar in Fig 3D, as both panels are calculated based on the case where the AP-2 number is 100 and the clathrin-clathrin binding rate is 50×0.913 μM⁻¹s⁻¹. For this system, increasing the AP-2 number does not increase the cluster number (fourth to last bars in Fig 3D), while increasing the clathrin-clathrin binding rate significantly increases the cluster number (the last two bars in Fig 4D), indicating that the clathrin-clathrin binding rate has a greater impact on cluster number than increasing the AP-2 number. In addition to the role of the clathrin-clathrin binding rate, we also explored the binding rate between AP-2 and clathrin and found that increasing this rate also leads to an increase in the number of clusters (S6 Fig). In conclusion, these results suggest that a high clathrin-clathrin binding rate leads to an increased cluster number, reduced cluster size, and shortened dwelling time.

### 3.4 Decreasing clathrin diffusion coefficient results in an increased cluster number, reduced cluster size, and shortened dwelling time

In the above analysis, we assumed that the surrounding environment of clathrin is water, which is less viscous than the cytosol. Therefore, the diffusion coefficient of clathrin in real cells [58,59] is usually smaller than the value used in the analysis. To investigate how the diffusion coefficient of clathrin affects the formation of multiple clusters, we next varied the clathrin diffusion coefficient while keeping other kinetic parameters constant. We found that, when the clathrin diffusion coefficient decreases, the most possible pattern exhibits a similar trend to those observed with increased $k_{(AP-2\cdot)Clat\cdot Clat}$: increased cluster number, decreased cluster size, and decreased dwelling time (Fig 5). Thus, a low clathrin diffusion coefficient results in an increased cluster number, reduced cluster size, and shortened dwelling time.

### 3.5 Simultaneous changes in the clathrin-clathrin binding rate and clathrin diffusion coefficient can reproduce the experimentally observed trend of FCLs under EGF stimulus

The above studies focus on clathrin behavior without stimulus, that is, all kinetic parameters in any given simulation are fixed. In this case, although both the clathrin number and size are dynamic after a long time simulation (e.g., after $t_c$), their mean behavior remains relatively stable, deviating from the experimentally observed increasing trend during the 15 minutes after the EGF stimulus. Therefore, we hypothesized that kinetic parameters associated with the assembly of FCL might change in response to an EGF stimulus. To test this hypothesis, we started from a state where there is only one cluster (Fig 6A), and then tested different combinations of parameter changes (Fig 6B) to determine which one best reproduces the experimental results.

First, we varied the kinetic parameters to explore which configurations could reproduce the experimentally observed clathrin dynamics during 30 minutes after the EGF stimulus. In this time interval, experimental data showed that both the

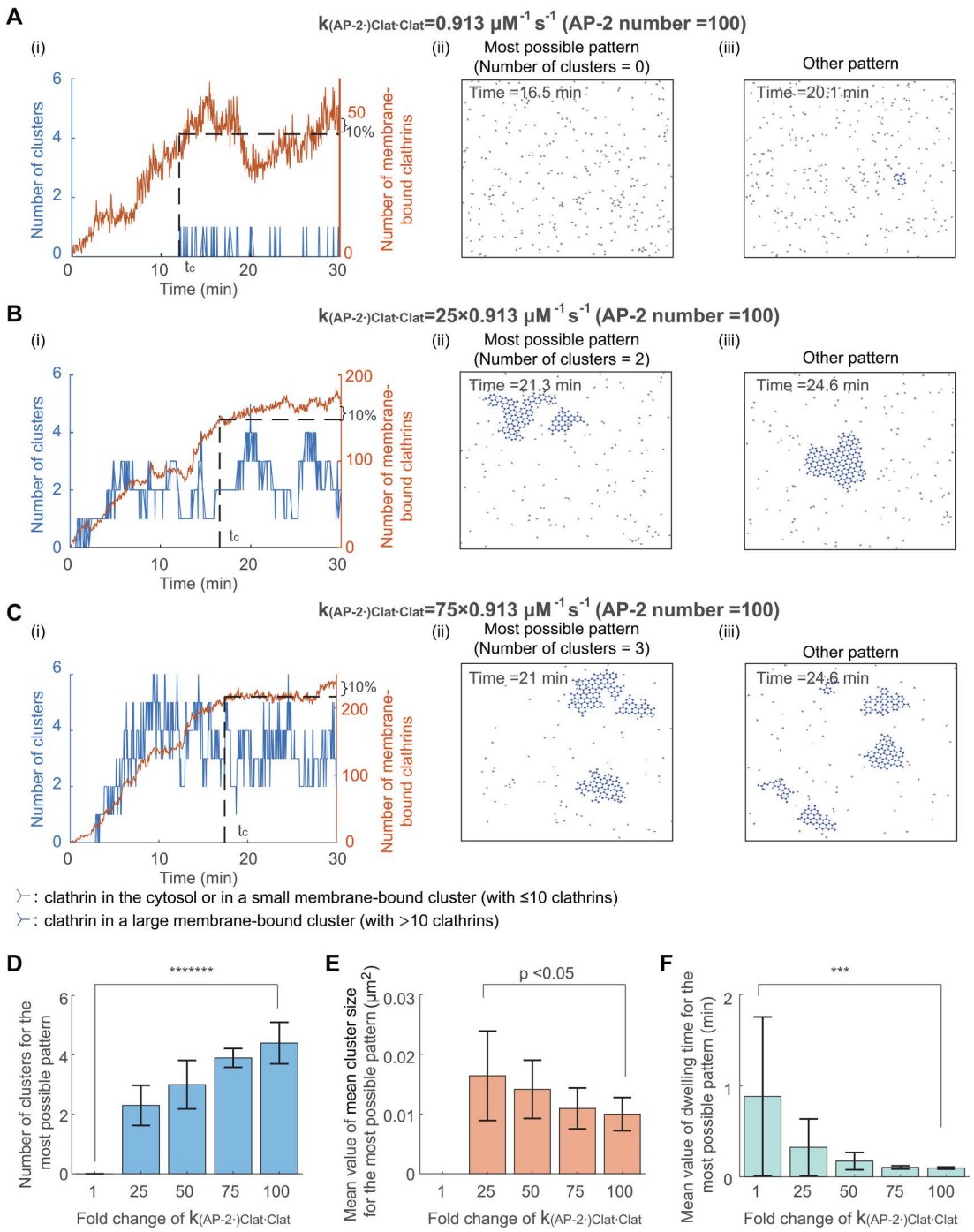

**Fig 4. Increasing clathrin-clathrin binding rate leads to an increased cluster number, reduced cluster size, and shortened dwelling time. (A-C)** Simulated clathrin dynamics when the clathrin-clathrin binding rate $k_{(AP-2\cdot)Clat\cdot Clat}$ is 0.913 $\mu M^{-1}s^{-1}$ **(A)**, 25 × 0.913 $\mu M^{-1}s^{-1}$ **(B)**, and 75 × 0.913 $\mu M^{-1}s^{-1}$ **(C)**. Parameters except $k_{(AP-2\cdot)Clat\cdot Clat}$ are fixed. The initial condition is that clathrins are randomly distributed in the simulation domain. Panels (i-iii) are plotted in the same way as those in Fig 3A. **(D)** The cluster number for the most possible pattern when increasing $k_{(AP-2\cdot)Clat\cdot Clat}$. **(E)** The mean value of the mean cluster size for the most possible pattern when increasing $k_{(AP-2\cdot)Clat\cdot Clat}$. **(F)** The mean dwelling time for the most possible pattern when increasing $k_{(AP-2\cdot)Clat\cdot Clat}$. P value was obtained from unpaired t-test: *** indicates $p<0.001$; ******* indicates $p<1E-7$. Data in (D-F) were shown as mean±SD, where SD means standard deviation.

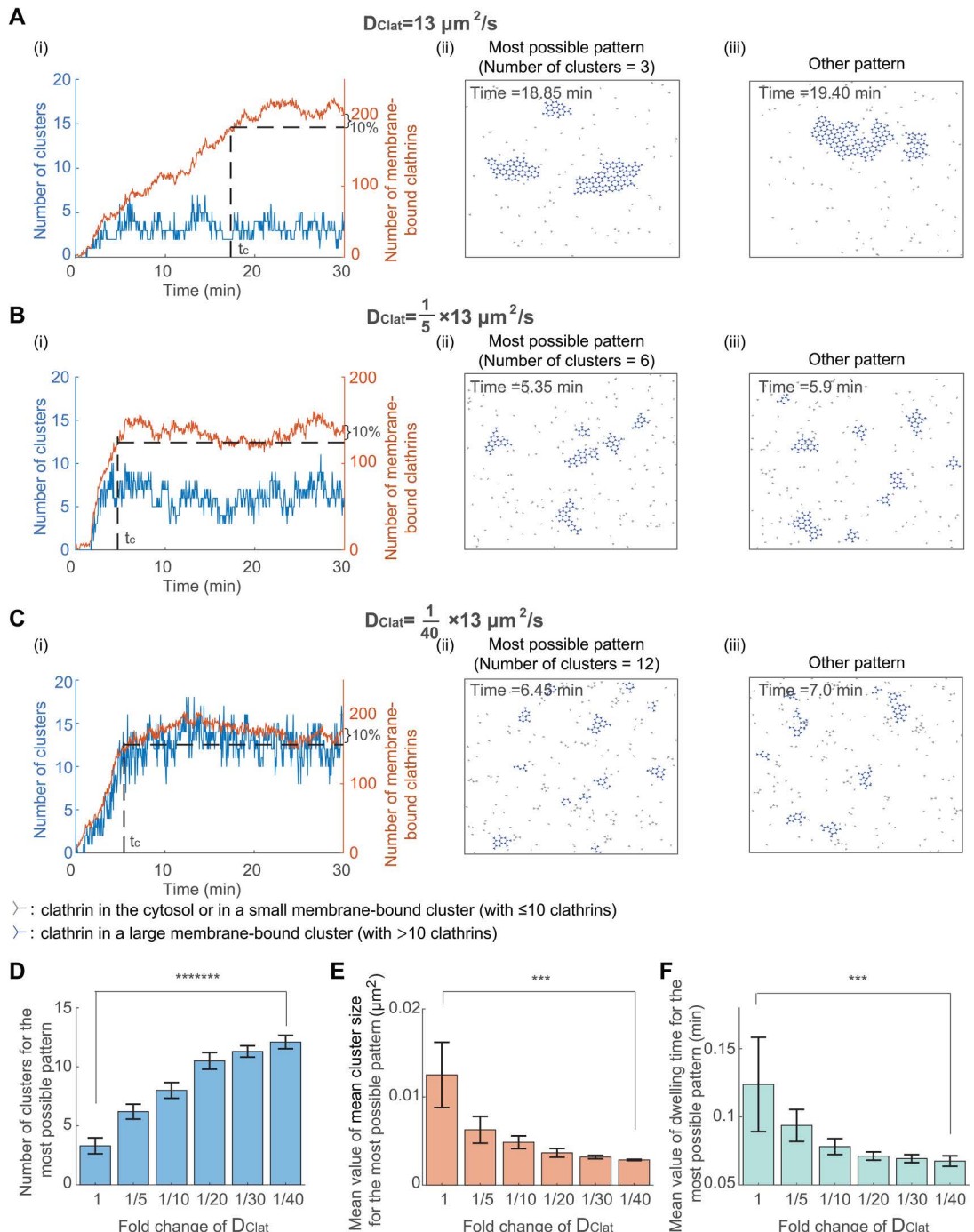

**Fig 5. Decreasing clathrin diffusion coefficient results in an increased cluster number, reduced cluster size, and shortened dwelling time.** (A-C) Simulated clathrin dynamics when the clathrin diffusion coefficient $D_{Clat}$ is 13 μm²/s [30] (A), $\frac{1}{5} \times 13$ μm²/s (B), and $\frac{1}{40} \times 13$ μm²/s (C). Parameters except $D_{Clat}$ are fixed. The initial condition is that clathrins are randomly distributed in the simulation domain. Panels (i-iii) are plotted in the same way as those in Fig 3A. (D) The cluster number for the most possible pattern when decreasing $D_{Clat}$. (E) The mean value of the mean cluster size for the most possible pattern when decreasing $D_{Clat}$. (F) The mean dwelling time for the most possible pattern when decreasing $D_{Clat}$. P value was obtained from unpaired t-test: *** indicates p<0.001;******* indicates p<1E-7. Data in (D-F) were shown as mean±SD, where SD means standard deviation.

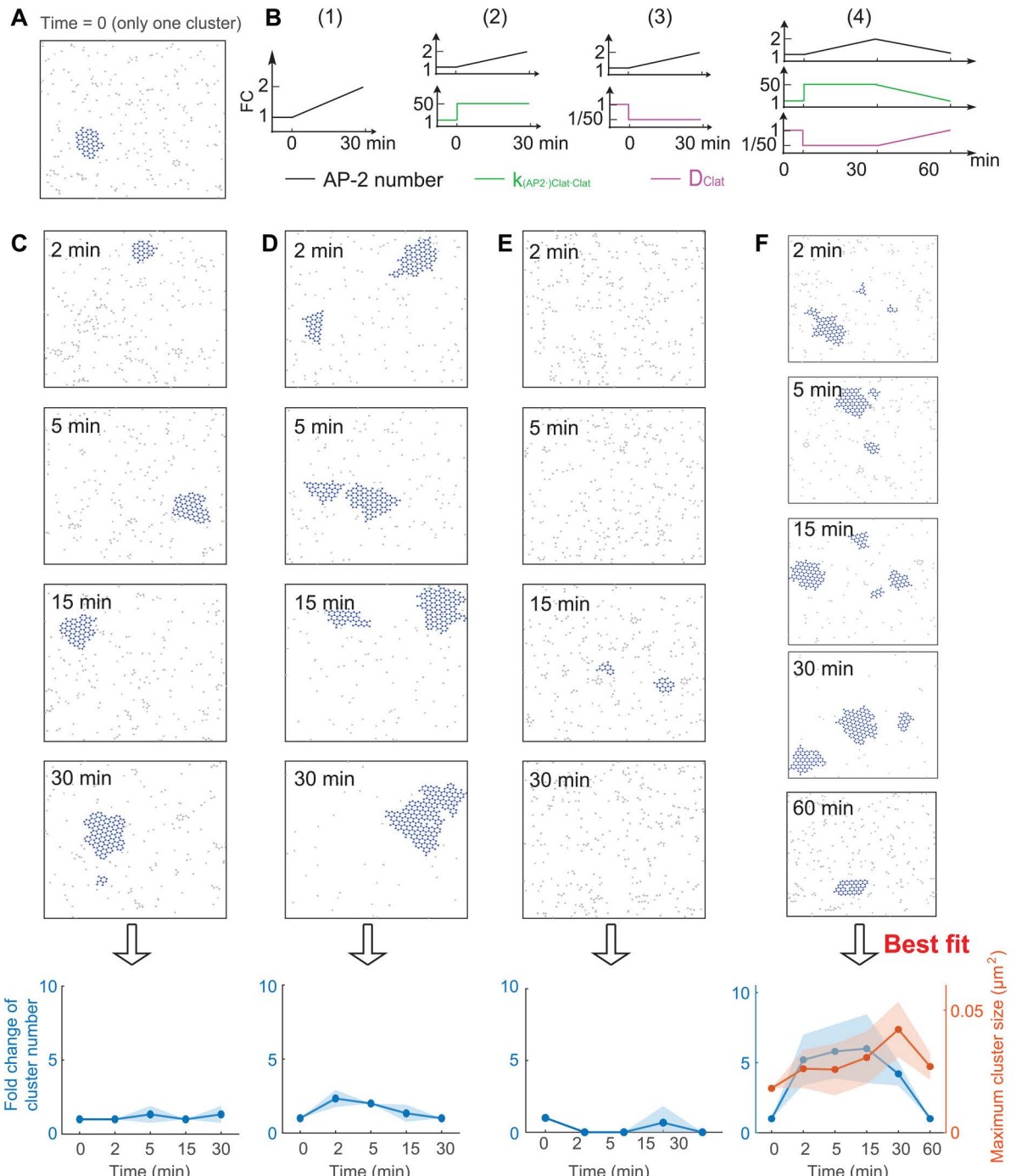

**Fig 6. Simultaneous changes in the clathrin-clathrin binding rate and clathrin diffusion coefficient can reproduce the experimentally observed trend of FCLs under EGF stimulus.** **(A)** Initial condition used to mimic the clathrin cluster before EGF stimulus. **(B)** Changing kinetic parameters to mimic the effect of EGF stimulus. Four different combinations were tested: (1) increasing AP-2 number only; (2) increasing AP-2 number and clathrin-clathrin binding rate simultaneously; (3) increasing AP-2 number and decreasing clathrin diffusion coefficient simultaneously; (4) increasing AP-2 number, increasing, and decreasing simultaneously. **(C–F)** Clathrin dynamics on the cell membrane under different parameter changes in **(B)**. The panels (C–F) correspond to parameter changes in (1–4) in the panel **(B)**, respectively. For each type of parameter change, four or five snapshots of clathrins at different time points were shown. The last column shows the fold change in cluster number (left) and maximum cluster size (right) as a function of time. For each type of parameter change, three replicates were performed, shown as mean SD (standard deviation). Solid lines in the last row of (C–F) denote the mean, with the shading representing the standard deviation.

cluster number and cluster size increase (Fig 1E and [16]). As shown in the previous sections, increasing AP-2 number alone, increasing clathrin-clathrin binding rate $k_{(AP-2\cdot)Clat\cdot Clat}$ alone, or decreasing clathrin diffusion coefficient $D_{Clat}$ alone all lead to the increased cluster number. The increase in clathrin-clathrin binding rate may result from EGF-induced phosphorylation of the clathrin heavy chain [60], while the decrease in $D_{Clat}$ may be attributed to EGF-triggered aggregation of various molecules, including β5 integrin and EGF receptors, into the clathrin cluster [16]. Thus, in order to generate the increased cluster number during 30 minutes after EGF stimulus, these three parameters might be changed, but it remains unknown whether these three parameters should be changed individually or simultaneously. Thus, we tested the following four ways of changing parameter (Fig 6B): (1) increasing the AP-2 number; (2) increasing the AP-2 number and increasing clathrin-clathrin binding rate $k_{(AP-2\cdot)Clat\cdot Clat}$ simultaneously; (3) increasing the AP-2 number and decreasing clathrin diffusion coefficient $D_{Clat}$ simultaneously; (4) increasing AP-2 number, increasing $k_{(AP-2\cdot)Clat\cdot Clat}$, and decreasing $D_{Clat}$ simultaneously. For all methods, AP-2 number is increased from 100 to 200 in the time interval [0, 30 minutes]; $k_{(AP-2\cdot)Clat\cdot Clat}$ is increased by a 50-fold change; $D_{Clat}$ is decreased to one-fiftieth of its previous value. We found that the clathrin dynamics in the first three methods do not show a significant increase in cluster number in the time interval [0, 30 minutes] (Fig 6C–6E). However, the fourth method, where AP-2 number, $k_{(AP-2\cdot)Clat\cdot Clat}$ and $D_{Clat}$ are all changed, exhibits an increased cluster number and increased cluster size in the time interval [0, 30 minutes] (Fig 6F). These results indicate that a simultaneous increase in AP-2 number and $k_{(AP-2\cdot)Clat\cdot Clat}$, along with a decrease in $D_{Clat}$, can generate the experimentally observed cluster dynamics during the first 30 minutes of the EGF stimulus.

After we reproduced the clathrin dynamics in the first 30 minutes of EGF stimulus, we hypothesized that reversing all the kinetics parameters would reproduce the experimentally observed clathrin disappearance between 30–60 min after EGF stimulation. We observed that decreasing AP-2 number, $k_{(AP-2\cdot)Clat\cdot Clat}$ and $D_{Clat}$ lead to decreases in both FCLs cluster number and size at 60 min of EGF stimulation (Fig 6F). This result is consistent with the experimental data collected after 60 minutes of EGF stimulation (Fig 1E). Between 30 and 60 minutes, numerical simulations predicted that large clusters break apart into many smaller clusters (S7 Fig) and thus result in a temporary increase in the number of clusters. These results also suggest that reverting the AP-2 number, $k_{(AP-2\cdot)Clat\cdot Clat}$, and $D_{Clat}$ to their baseline values reproduces the clathrin dynamics during 30–60 minutes after EGF stimulus.

## 4 Discussion

Clathrin can assemble into various shapes, including Ω-shaped pits and flat clathrin lattices. The Ω-shaped pits that are formed on the cell membrane during endocytosis are short-lived and are removed from the membrane after the process. In contrast, flat clathrin lattices tend to be more stable, lasting more than 10 minutes, and have different functions compared to the Ω-shaped pits. These flat clathrin lattices have been found to act as signaling hubs. For example, their number and size exhibit an increasing trend followed by a decreasing trend after the EGF stimulus. However, how such changes in number and size are achieved remains elusive. Here, we investigated the mechanisms underlying the formation of multiple FCLs, as well as the changes in FCLs induced by EGF stimulus.

We used a particle-based model for clathrins and simulated the dynamics using NERDSS, a nonequilibrium simulator for multibody self-assembly [31]. For the mechanism of multiple FCLs formation, we started from a random distribution of clathrins and only focused on the clathrin state when the system becomes "stable," that is, the time when the number of membrane- bound clathrins reaches 90% of that at the end of the simulation. We revealed that the FCL formation is dynamic, suggested by the highly dynamic cluster number and cluster size. Moreover, when increasing AP-2 number, clathrin states exhibit a transition from no cluster to short-lived multiple small clusters, and finally to long-lasting single giant cluster. An increase in the clathrin-clathrin binding rate or a decrease in the clathrin diffusion coefficient both lead to an increased cluster number, a decreased cluster size, and reduced dwelling time. In addition to the mechanism of multiple FCL formation, we also studied the mechanisms of changes in FCLs caused by EGF stimulus. We found that the changes in AP-2 number are not sufficient to achieve the experimentally observed trend of FCLs. However, when

combined with changes in clathrin-clathrin binding rate and clathrin diffusion coefficient, the simulated clathrin dynamics qualitatively fit the experimental data.

The cluster size and number obtained from the simulations are broadly consistent with the experimental data. In [45], the cluster sizes ranged from 0.01 $\mu m^2$ to 0.3 $\mu m^2$ in HeLa and U87 cells, and from 0.01 $\mu m^2$ to 1.3 $\mu m^2$ in MCF7 cells (also see Fig 1C). Despite the presence of very large clusters, the mean cluster size was near 0.05 $\mu m^2$ across all three cell types. In our simulations, the cluster sizes ranged from 0.001 $\mu m^2$ to 0.1 $\mu m^2$, which is close to the mean cluster size observed in experimental data. Moreover, the cluster number in 1 $\mu m \times$ 1 $\mu m$ square ranges from 0 to 10 in HeLa, U87, and MCF7 cells, with 1 or 2 clusters occurring most frequently ([45] and Fig 1D). In this study, we also simulated clathrin dynamics on a 1 $\mu m \times$ 1 $\mu m$ cell membrane region and observed 0–12 clusters, consistent with the experimentally reported cluster numbers. As for dwelling time, by measuring the intensity of fluorescence-labeled clathrin, Grove et al. found that flat clathrin lattices are long-lived, typically lasting more than 10 minutes [7]. However, this technique cannot distinguish small individual flat clathrin lattices. In contrast, the electron microscopy technique used in [16] has a better than fluorescence imaging and observe small individual flat clathrin lattices, but it cannot monitor cluster dynamics because cells are fixed during imaging. Therefore, large flat clathrin lattices may have long dwelling times more than 10 minutes, but it remains unclear whether small lattices exhibit short or long dwelling times. In our simulations, the dwelling time ranges from 0.05 to 10 minutes depending on cluster size, which need to be validated by future experiments.

The roles of kinetic parameters in multiple FCLs formation found in this work improve our understanding of clathrin assemblies. We found that increasing clathrin-clathrin binding rate leads to an increased cluster number, reduced cluster size, and shortened dwelling time. Such an increase in the number of clusters may be because a high value of $k_{(AP\text{-}2 \cdot Clat \cdot Clat)}$ allows formed clusters to easily absorb unbound clathrins when the component occasionally dissociates. Furthermore, as more clusters arise, they compete for the limited clathrin resources, potentially resulting in a decrease in cluster size. As small clusters do not have too many clathrins and may lose all their clathrins in a very short time, small clusters tend to disappear quickly, which may cause a decrease in dwelling time.

As for the role of clathrin diffusion coefficient, we showed that decreasing clathrin diffusion coefficient results in an increased cluster number, reduced cluster size, and shortened dwelling time. This type of behavior was also observed in recent work [61]. However, these results seem to deviate from the property for the steady-state system, where the distribution of clusters is expected to remain unchanged. The possible reason might be that a slower clathrin diffusion coefficient prevents dissociated clathrin from dispersing too far, thus allowing the clathrin clusters to efficiently recruit the dissociated clathrin and maintain their structure, rather than merging into a giant cluster or disassembling. Thus, the effect of the diffusion coefficient on clathrin dynamics could be an artifact of the algorithm, as the phenomenological acceptance criteria can lead to a low acceptance rate for two clusters to combine when molecule diffusion becomes slow.

The model predictions are consistent with previous biochemical evidence to some extent. On the one hand, we predicted that the increase in the AP-2 number helps FCLs grow after EGF stimulus. This is closely related to the experimentally observed increased interaction between clathrin and endocytic adaptors (e.g., AP-2, Eps15) after EGF stimulus [62–64]. Moreover, there is a regulatory domain on the EGFR cytoplasmic domain that is essential for AP-2 recruitment [65]. On the other hand, we predicted that the change in clathrin-clathrin binding rate might be important for EGF-triggered FCLs dynamics. Such changes may relate to EGF-triggered phosphorylation of AP-2 and clathrin [60,66]. Further experimental work is necessary to validate this hypothesis. However, it should be noted that other possible conditions may also generate the experimentally observed EGF-triggered cluster dynamics, such as changes in the clathrin-clathrin unbinding rate or the binding rate between AP-2 and clathrin, which were not tested in this study. Future detailed simulations will be necessary to explore all the potential mechanisms underlying the EGF-triggered FCL changes.

The way we altered the kinetic parameters in Fig 6B (iii) is closely related to the activity of EGFR pathway. Upon persistent 60-minute EGF stimulation, EGFR and downstream kinases are first activated and then deactivated [67]. At the same time, some downstream molecules also exhibit an enrichment followed by degradation, returning to pre-stimulus

levels, for example, Grb2 [68]. Thus, we predict that these changes in EGFR and downstream pathways affect the AP-2 number, clathrin-clathrin binding rate, and clathrin diffusion coefficient, that is, all of these parameters exhibit a sudden change at 30 minutes (as shown in Fig 6B (iii)). Here, the change in AP-2 number can be interpreted as a change in the available AP-2, since the the EGFR pathway may regulate the state of AP-2. In addition, upon EGF stimulation, EGFR endocytosis is triggered on the cell membrane. This may cause the FCL to form Ω-shaped pits that leave from the membrane, leading to a reduction in FCL on the surface. This endocytosis process may contribute to the decrease in FCL number and area between 30 and 60 minutes following EGF stimulation. Further experiments are needed to test these hypotheses.

FCLs regulate various cellular processes like proliferation and actin network remodeling [9,69]. FCLs also harbor different receptors including epidermal growth factor receptor (EGFR), fibroblast growth factor receptor 1 (FGFR1), low-density lipopoprotein receptor LDLR, lysophosphatidic acid receptor 1 (LPAR1), and β5 integrin [16,70]. β 5-integrin knock down leads to decreased abundance of FCLs at the plasma membrane [16]. Therefore, flat clathrin lattices serve as multifunctional platforms that coordinate signaling, endocytosis, and cytoskeletal organization. Despite the distinct dynamic behaviors of FCLs under various stimuli, their assembly and disassembly may universally be highly dynamic.

In addition to the particle-based model we used in this work, lattice model [71,72], Cahn–Hilliard equation [73–75] Turing model [76,77], molecular dynamics simulations [78,79] have also been used to model the protein clusters within the cell. However, the continuous models, including Cahn–Hilliard equation and Turing model, do not include the stochasticity of chemical reactions, and thus may lead to the failure of capturing experimental observations. This failure was also validated by simulating Turing model (S8 Fig, S1 Text, S3 Table), where the increases in cluster number and size occur at different times. In addition to this shortcoming, the assumption of the Turing model may also be unrealistic for the clathrin-AP-2 system. Specifically, the Turing model requires that AP-2, as an activator, diffuses more slowly than the membrane-bound clathrin. However, this may not hold true in the clathrin-AP-2 system, as the membrane-bound clathrin is a complex composed of both AP-2 and clathrin, which could cause it to diffuse more slowly than free AP-2. Thus, when simulating the dynamics of multiple clusters, continuous models may not be able to fit experiments as well as particle-based models.

In this work, we focused on the interactions between AP-2 and clathrins with several simplifications. For example, there are multiple AP-2 binding sites on clathrin [80], but we assume only one AP-2 binding site per cluster for the sake of simplicity; the binding between AP-2 and the cell membrane is neglected by assuming that AP-2 is located on the cell membrane at all times. To ensure that the reducing the clathrin-AP-2 binding sites not quantitatively affect the result, we tested effect of AP-2 number (S9 Fig) and clathrin-clathrin binding sites (S10 Fig) when there are three clathrin-AP-2 binding sites. Though the exact number of clusters, cluster size, and dwelling time are different between the systems with one and three clathrin-AP-2 binding sites, increasing the number of AP-2 exhibits a similar trend (panels D-F in S9 Fig): the transition from no clusters to short-lived, multiple small clusters, and then to a long-lasting, single giant cluster. In addition, in the system with three clathrin-AP-2 binding sites, increasing the clathrin-clathrin binding rate leads to an increase in the number of clusters (panels D-F in S10 Fig), which is consistent with the result observed when there is only one clathrin-AP-2 binding site. Therefore, we assume that the simplification from three clathrin-AP-2 binding sites to one only qualitatively affects the results. In addition, we assumed a fixed number of 300 clathrin molecules, neglecting their production and degradation. As a result, the number of membrane-bound clathrins cannot exceed 300. Given the complexity of the protein synthesis process, we anticipated that clathrin production could influence the results over longer timescales. Another limitation is that we did not consider the reactions between clathrins and signaling molecules involved in EGFR pathway. In this study, we predicted that, to ensure the increasing trend of FCL number and size in the first 30 minutes and then decreasing trend between 30 and 60 minutes, EGFR pathway will regulate the AP-2 number, clathrin-clathrin binding rate, and clathrin diffusion coefficient simultaneously. However, the mechanism of how EGFR pathway regulates these parameters remains unclear. Future work is expected to include both EGFR pathways and clathrins to investigate

the crosstalk between the EGFR pathway and FCLs. Furthermore, there are debates about the relationship between flat clathrin lattices and Ω-shaped pits [2,11,81,82], which were not considered in this work. Both constant curvature and constant area models have been used to explain the transition from flat to curved clathrin assemblies [11]. To further investigate the crosstalk among flat clathrin lattices, Ω-shaped pits, and EGFR pathways, which still remain elusive, future modeling studies will need to develop a model that incorporates membrane properties, self-assembled clathrins, and EGFR signaling pathways.

In addition to the flat structural assemblies formed by clathrin examined in this work, other proteins can also form assemblies or clusters with distinct shapes within the cell, including nanodomain formed by A-kinase anchoring protein (AKAP) 79/150 [83], stress granule [84], and Yes-associated protein (YAP) [85] condensates. Researchers have extensively explored the formation mechanisms of these protein assemblies [86,87]. For example, increased monomer concentration leads to the transition from co-existing multiple assemblies to one giant assembly [71], consistent with our simulation results when increasing AP-2 number; the lag time and steepness for the clathrin cluster growth curve are primarily controlled by the individual clathrin binding rate to membrane and the binding rate between clathrin and AP-2, respectively [30]; two-dimensions mesoscopic simulations allow the formation of multiple postsynaptic protein assemblies while three-dimensions cannot [88]. In addition to formation mechanisms, the interaction between protein assemblies and neighboring environment also has been widely studied [87,89,90]. For instance, protein assemblies not only regulate key signaling pathways [91,92] but also affect the kinetic parameters [93]; curvature-inducing proteins can assemble and regulate cell membrane shape [94]. This study also serves as an example for investigating the interactions between protein assemblies and their surrounding environment.

## Supporting information

**S1 Text. The Turing model cannot simultaneously achieve the increase of cluster size and the increase of cluster number.**
(PDF)

**S1 Table. Parameters used in the particle-based model (from [30]).**
(PDF)

**S2 Table. Relative coordinate for clathrin.**
(PDF)

**S3 Table. Parameters used in Turing model.**
(PDF)

**S1 Fig. The FCL distributions in MCF7 cells.** (A–C) The same plots as those in Fig 1B–1D except the cell type. (D) Same plot as in Fig 1E, except that only 1 μm × 1 μm square areas are considered instead of the entire cell membrane. Here, we focus only on the neighboring 1 μm × 1 μm square areas surrounding each cluster.
(PDF)

**S2 Fig. The number of clusters as a function of time for different values of $k_{Clat-Clat}$.** $k_{Clat-Clat}$ represents the clathrin-clathrin binding rate when neither clathrin is bound to AP-2. Excecpt the difference in $k_{Clat-Clat}$, the settings of model simulations are the same as that in Fig 4C.
(PDF)

**S3 Fig. Two snapshots showing clathrins and AP-2 molecules on the cell membrane.** (A) The distribution of clathrin (blue or gray) and AP-2 (red) at 30 minutes, simulated from an initial condition where the molecules are randomly distributed. Clathrin is shown in blue or gray, depending on whether it is membrane-bound. 45 clathrin-bound AP-2 molecules

are labeled in red, while the remaining 55 AP-2 molecules are not shown. (B) Same plot as (A), but with a different AP-2-clathrin binding rate. In this case, 95 clathrin-bound AP-2 molecules are shown in red, while the remaining 5 AP-2 molecules are not shown. See S1 Table for kinetic parameters.
(PDF)

**S4 Fig. Distributions of cluster size and dwelling time in** Fig 2. (A) Distributions of mean cluster size (upper panel) and dwelling time (lower panel) for all data points, obtained from the simulations shown in Fig 2. (B–E) Same plots as in (A), but restricted to data points with cluster numbers of 1 (B), 2 (C), 3 (D), and 4 (E), respectively. (F) The mean value of the cluster size in panels (B–E). $S_1$ represents the mean cluster size in panel (B), corresponding to the highest dashed line. The dashed lines, from the second highest to the lowest, represent $\frac{1}{2}S_1$, $\frac{1}{3}S_1$, and $\frac{1}{4}S_1$, respectively. (G) Standard deviation of the mean cluster size in panels (A–E).
(PDF)

**S5 Fig. The FCL exhibits the phase transition from no cluster to one giant cluster when the AP-2 number increases while maintaining a low clathrin-clathrin binding rate.** (A–F) The same plots as those in Fig 3 except that a low clathrin-clathrin binding rate is used.
(PDF)

**S6 Fig. Increasing the AP2-clathrin binding rate leads to an increase in cluster number.** (A–B) Simulated clathrin dynamics when the AP2-clathrin binding rate $k_{AP\text{-}2\cdot Clat}$ is $0.5 \times 0.0012\ \mu M^{-1}s^{-1}$ (A), $10 \times 0.0012\ \mu M^{-1}s^{-1}$ (B) Parameters except $k_{AP\text{-}2\cdot Clat}$ are fixed. The initial condition is that clathrins are randomly distributed in the simulation domain. In the left panel, the number of clusters (in blue) and the number of membrane-bound clathrins (in red) are shown as a function of time. Panels on the right showed the most possible pattern. (C) The cluster number for the most possible pattern when increasing $k_{AP\text{-}2\cdot Clat}$. Data were shown as mean±SD, where SD means standard deviation.
(PDF)

**S7 Fig. Simulated FCL dynamics after 30 minutes of EGF stimulus, where kinetic parameters change as shown in Fig 6B(4).**
(PDF)

**S8 Fig. The Turing model cannot simultaneously achieve the increase of cluster size and the increase of cluster number.** (A) Schematic of reactions between the AP-2 and the clathrin. The AP-2 improves the recruitment of itself and the clathrin to the cell membrane. Once bound to the cell membrane, the clathrin inhibits the accumulation of the AP-2 on the cell membrane due to steric repulsion. (B) The snapshots of clathrin clusters after increasing the association rate of AP-2 and membrane $\mu$ by 80% at time 0. The first plot corresponds to the Turing pattern with parameters in S3 Table. Scale bar is 1 μm. (C) The size for each clathrin cluster (left axis) and the total number of clathrin clusters (right axis) at the time points in (B). The increase in the clathrin cluster size only occurs between 0 and 15 seconds, while the increase of the total number of clathrin clusters occurs after 15 seconds. (D) The same plot as that in B, except that the AP-2 concentration is shown instead of the clathrin concentration.
(PDF)

**S9 Fig. Effect of AP-2 number when there are 3 clathrin-AP-2 binding sites.** (A–C) Same plots as Fig 3A–3C, except with three clathrin-AP-2 binding sites instead of one. Here, only the most possible pattern was shown, while other patterns were not shown. (D–F) Same plots as Fig 3D–3F, but with the results for one and three clathrin-AP-2 binding sites plotted.
(PDF)

**S10 Fig. Effects of the clathrin-clathrin binding rate when there are 3 clathrin-AP-2 binding sites.** (A–C) Same plots as Fig 4A–4C, except with three clathrin-AP-2 binding sites instead of one. Here, only the most possible pattern was

shown, while other patterns were not shown. (D–E) Same plots as Fig 4D–4E, but with the results for one and three clathrin-AP-2 binding sites plotted. (F) Example of the locations of clathrin (blue) and AP-2 (red). (PDF)

## Author contributions

**Conceptualization:** Justin W. Taraska, Padmini Rangamani.

**Data curation:** Marco A. Alfonzo-Méndez, Justin W. Taraska.

**Formal analysis:** Marco A. Alfonzo-Méndez.

**Methodology:** Lingxia Qiao.

**Resources:** Lingxia Qiao, Marco A. Alfonzo-Méndez, Justin W. Taraska, Padmini Rangamani.

**Software:** Lingxia Qiao, Padmini Rangamani.

**Supervision:** Justin W. Taraska, Padmini Rangamani.

**Visualization:** Lingxia Qiao, Marco A. Alfonzo-Méndez.

**Writing – original draft:** Lingxia Qiao, Marco A. Alfonzo-Méndez, Justin W. Taraska, Padmini Rangamani.

**Writing – review & editing:** Lingxia Qiao, Marco A. Alfonzo-Méndez, Justin W. Taraska, Padmini Rangamani.

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
