## [Decision Letter · Decision Letter 0]

20 Jul 2025

Dynamics of the formation of flat clathrin lattices in response to growth factor stimulus

PLOS Computational Biology

Dear Dr. Rangamani,

Thank you for submitting your manuscript to PLOS Computational Biology. After careful consideration, we feel that it has merit but does not fully meet PLOS Computational Biology's publication criteria as it currently stands. Therefore, we invite you to submit a revised version of the manuscript that addresses the points raised during the review process.

Please submit your revised manuscript within 60 days Sep 19 2025 11:59PM. If you will need more time than this to complete your revisions, please reply to this message or contact the journal office at ploscompbiol@plos.org. Please include the following items when submitting your revised manuscript:

We look forward to receiving your revised manuscript.

Kind regards,

Rotem Rubinstein, PhD

Guest Editor

PLOS Computational Biology

Nir Ben-Tal

Section Editor

PLOS Computational Biology

**Additional Editor Comments:**

Please note that the reviewers (especially reviewer 2) have raised important concerns, and it is important that these be addressed satisfactorily before the manuscript can be accepted for publication.

**Journal Requirements:**

At this stage, the following Authors/Authors require contributions: Lingxia Qiao, Marco A. Alfonzo-Méndez, Justin W. Taraska, and Padmini Rangamani. Please ensure that the full contributions of each author are acknowledged in the "Add/Edit/Remove Authors" section of our submission form.

4) Your manuscript is missing the following sections: Methods.  Please ensure all required sections are present and in the correct order. Make sure section heading levels are clearly indicated in the manuscript text, and limit sub-sections to 3 heading levels. An outline of the required sections can be consulted in our submission guidelines here:

5) Please upload all main figures as separate Figure files in .tif or .eps format. For more information about how to convert and format your figure files please see our guidelines:

6) We notice that your supplementary Figures, and Tables are included in the manuscript file. Please remove them and upload them with the file type 'Supporting Information'. Please ensure that each Supporting Information file has a legend listed in the manuscript after the references list.

7) Please note that your Data Availability Statement is currently missing the repository name, and the DOI/accession number of each dataset OR a direct link to access each dataset. If your manuscript is accepted for publication, you will be asked to provide these details on a very short timeline. We therefore suggest that you provide this information now, though we will not hold up the peer review process if you are unable.

8) Please amend your detailed Financial Disclosure statement. This is published with the article. It must therefore be completed in full sentences and contain the exact wording you wish to be published.

9)  Please ensure that the funders and grant numbers match between the Financial Disclosure field and the Funding Information tab in your submission form. Note that the funders must be provided in the same order in both places as well.

**Reviewers' comments:**

Reviewer's Responses to Questions

**Comments to the Authors:**

Reviewer #1: Multiple flat clathrin lattices exist on the cell membrane serving as signaling hubs after epidermal growth factor stimulation. This paper used a particular-based model to simulate the assembly and disassembly of flat clathrin lattices to capture these dynamics. The results are quite interesting and could be useful to reveal the mechanisms underlying the formation of flat clathrin lattices and how epidermal growth factor regulate their dynamics. I suggest the authors to consider the following comments and incorporate the corresponding revisions before the manuscript can be accepted for publication.

1. In the introduction section, the authors should also mention a little bit background about the application of particle-based simulations to other membrane protein systems, such as TNF-induced receptor clustering by Su et al.

2. In the model development part, there are several issues needed to be clarified. First, whether periodic boundary conditions were applied along x and y directions? How about the direction that is perpendicular to the cell membrane?

3. How to describe the 2D diffusion of AP-2 in the cell membrane? How to determine its 2D diffusion coefficient in the living cell?

4. The simulations can directly give the cluster number, cluster size and dwelling time as measurable quantities. If any of these are comparable to the available experimental data?

5. In the results part 3.3, the authors adjusted the binding rate between clathrins. Is this the association rate or dissociation rate?

6. It would be interesting to see how the dynamics of assembly and disassembly would be regulated by changing the binding rate between clathrin and AP-2.

7. In 3.5, the authors investigated the trend of clathrin lattices under EGF stimulus. One concern resulted from the fact that only four scenarios were tested and the authors argued that the last combination can lead to the experimentally observed trend. This, however, cannot exclude other possible conditions which might also lead to the similar behavior of clathrin lattice assembly.

Reviewer #2: The authors present a simulation study on the number and average size of flat clathrin lattices, exploring the effects of three parameters in particular. These three parameters are then used to explain the experimentally observed growth of lattices, and their subsequent return to the equilibrium state, when the cell is stimulated with epidermal growth factor. The study shows some interesting trends, but also raises a number of questions.

The authors reduce the number of clathrin-AP2 binding sites to one, which "we assume that it does not quantitatively affect the results" (p5). But, since this reduction rules out cooperative effects of clathrin binding to multiple AP2, it may even affect the results qualitatively.

The peculiar addition rule for diffusion coefficients, page 5, has its consequences. The translation diffusion coefficient of a spherical cluster scales inversely proportional with its radius (Stokes-Einstein), not with the number of monomers. Its rotation diffusion diffusion coefficient scales inversely proportional with the radius cubed (Stokes-Einstein-Debye), not with the number of monomers cubed. Consequently, the simulated clusters diffuse too slowly (which following section 3.4 affects flat lattice formation).

Why are average cluster size and dwelling time calculated only for systems containing the most probable number of clusters? Why introduce this bias, in stead of analyzing all frames (after the equilibration period t_c)?

"we used a higher clathrin-clathrin binding rate than that in [30]" (p7). The notation "20 x 1.83" in table S1 suggests a 20 fold higher binding rate, but the value listed in table 1 of ref 30 is 0.083. Please specify the actual multiplication factor and a justification thereof.

In sections 3.4 and 3.5 the authors vary the diffusion coefficient of clathrin, reducing it by as much as a factor 40. What is the physical motivation for this variation? Does the triskelion change shape or does the cytosol viscosity change when a cell is exposed to EGF? Since the diffusion coefficient of a membrane-bound lattice is dominated by the number of membrane anchors, an increase in the number of AP2s would be a more natural explanation of a reduced lattice diffusion coefficient.

Minor comments:

* The abbreviation FCL is used in the abstract without proper introduction.

* Since reference 30 contains the original simulation model, I googled the title -- which turned out to be a poster abstract in BPJ. The provided PLOS page number is not correct either; the doi number works.

Reviewer #3: See attachment

**Have the authors made all data and (if applicable) computational code underlying the findings in their manuscript fully available?**

Reviewer #1: Yes

Reviewer #2: Yes

Reviewer #3: Yes

PLOS authors have the option to publish the peer review history of their article (what does this mean? ). If published, this will include your full peer review and any attached files.

**Do you want your identity to be public for this peer review?** For information about this choice, including consent withdrawal, please see our Privacy Policy .

Reviewer #1: **Yes:** Yinghao Wu

Reviewer #2: No

Reviewer #3: No

**Figure resubmission:**
---

## [Decision Letter · Decision Letter 1]

12 Feb 2026

Dear Dr. Rangamani,

We are pleased to inform you that your manuscript 'Dynamics of the formation of flat clathrin lattices in response to growth factor stimulus' has been provisionally accepted for publication in PLOS Computational Biology.

Best regards,

Nir Ben-Tal

Section Editor

PLOS Computational Biology

Reviewer's Responses to Questions

**Comments to the Authors:**

Reviewer #1: The revision has addressed all my comments; I recommend its publication on Plos Computational Biology.

Reviewer #2: I am satisfied with the revisions. It is hopefully clear to the readers, and authors, that the assumption regarding the addition of radii to calculate a diffusion coefficient is unrealistic.

Reviewer #3: The authors have addressed all of my comments and concerns.

**Have the authors made all data and (if applicable) computational code underlying the findings in their manuscript fully available?**

Reviewer #1: Yes

Reviewer #2: Yes

Reviewer #3: Yes

PLOS authors have the option to publish the peer review history of their article (what does this mean? ). If published, this will include your full peer review and any attached files.

**Do you want your identity to be public for this peer review?** For information about this choice, including consent withdrawal, please see our Privacy Policy .

Reviewer #1: **Yes:** Yinghao Wu

Reviewer #2: No

Reviewer #3: No

---

## [Editor Report · Acceptance letter]

PCOMPBIOL-D-25-01040R1

Dynamics of the formation of flat clathrin lattices in response to growth factor stimulus

Dear Dr Rangamani,

I am pleased to inform you that your manuscript has been formally accepted for publication in PLOS Computational Biology. Your manuscript is now with our production department and you will be notified of the publication date in due course.

With kind regards,

Anita Estes
